# Spatio-Temporal Variation of Groundwater Quality and Source Apportionment Using Multivariate Statistical Techniques for the Hutuo River Alluvial-Pluvial Fan, China

**DOI:** 10.3390/ijerph17031055

**Published:** 2020-02-07

**Authors:** Qianqian Zhang, Long Wang, Huiwei Wang, Xi Zhu, Lijun Wang

**Affiliations:** 1Hebei and China Geological Survey Key Laboratory of Groundwater Remediation, Institute of Hydrogeology and Environmental Geology, Chinese Academy of Geological Sciences, Shijiazhuang 050061, China; zhangqianqian@mail.cgs.gov.cn (Q.Z.); whuiwei@mail.cgs.gov.cn (H.W.); wlijun@mail.cgs.gov.cn (L.W.); 2School of Geology and Mining Engineering, Xinjiang University, Yanan Road, Urumqi 830047, China; wanglikang@mail.cgs.gov.cn

**Keywords:** groundwater quality, spatial variation, temporal variation, source apportionment, multivariate statistical techniques

## Abstract

Groundwater quality deterioration has become an environmental problem of widespread concern. In this study, we used a water quality index (WQI) and multivariate statistical techniques to assess groundwater quality and to trace pollution sources in the Hutuo River alluvial-pluvial fan, China. Measurement data of 17 variables in 27 monitoring sites from three field surveys were obtained and pretreated. Results showed that there were 53.09% of NO_3_^−^, 18.52% of SO_4_^2^^−^ and 83.95% of total hardness (TH) in samples that exceeded the Grade III standard for groundwater quality in China (GB/T 14848-2017). Based on WQI results, sampling sites were divided into three types: high-polluted sites, medium-polluted sites and low-polluted sites. The spatial variation in groundwater quality revealed that concentrations of total dissolved solids (TDS), Cl^−^, TH and NO_3_^−^ were the highest in high-polluted sites, followed by medium-polluted and low-polluted sites. The temporal variation in groundwater quality was controlled by the dilution of rainwater. A principal component analysis (PCA) revealed that the primary pollution sources of groundwater were domestic sewage, industrial sewage and water–rock interactions in the dry season. However, in the rainy and transition seasons, the main pollution sources shifted to domestic sewage and water–rock interactions, nonpoint pollution and industrial sewage. According to the absolute principal component scores-multivariate linear regression (APCS-MLR), most water quality parameters were primarily influenced by domestic sewage. Therefore, in order to prevent the continuous deterioration of groundwater quality, the discharge of domestic sewage in the Hutuo River alluvial-pluvial fan region should be controlled.

## 1. Introduction

The availability of good-quality groundwater is vital for the physical health and socio-economic development of the local populations that depend on it. Nowadays, the rapid growth in human populations and economic development has caused the severe deterioration of groundwater quality [1], especially in developing countries [2,3,4].

Groundwater quality is mainly controlled by two sets of factors: one is anthropogenic activity factors, such as domestic and industry sewage, agriculture fertilizer, urban development, exploitation of water resources and mining operations [4,5,6], while the other comprises natural processes, including aquifer lithology, soil characteristics, and groundwater–rock interaction [1]. According to Qin et al. [1], the major factors controlling groundwater quality in the coastal alluvial aquifers of the lower Liaohe River Plain, China were Holocene transgression and mixing, surface water infiltration, multi-factor processes, rainfall regimes and agricultural fertilizer contamination and geogenic fluoride enrichment. Herojeet et al. [7] recently demonstrated that groundwater chemistry in an alluvial aquifer of Nalagarh Valley, India was strongly controlled by water–rock interaction, ion exchange and leaching of parent materials, agricultural runoff and the seepage of industrial and domestic wastes material.

An effective way to mitigate and control the continuous deterioration of groundwater quality is to understand the spatial and temporal variations and identify the major pollution sources, which requires continuous and regular water quality monitoring programs. However, such monitoring systems generate voluminous databases and their analysis requires robust analytical tools [8]. In recent years, multivariate statistical techniques (such as principal component analysis (PCA), factor analysis (FA) and absolute principal component score-multiple linear regression (APCS-MLR)) have been widely used in the water environment to evaluate both temporal and spatial variations of water quality [8,9,10], to provide qualitative information about potential pollution sources [6,11,12,13,14,15], and to estimate source distributions for each pollution variable [4,11].

The Hutuo River alluvial-pluvial fan is located in the western portion of the North China Plain. Nowadays, with rapid urbanization and industrialization, the groundwater quality has been seriously affected by anthropogenic activities [16]. A previous study found that groundwater quality in the Hutuo River region was primarily affected by inorganic materials including total hardness, iron, manganese, total dissolved solids and nitrate nitrogen [17]. Zhang et al. [4] found that the poor groundwater quality in the Hutuo River region is due to intense anthropogenic activities as well as aquifer vulnerability to contamination. In this study, we mainly distinguish the temporal and spatial patterns of groundwater quality, evaluate the spatiotemporal variation in groundwater quality using a water quality index (WQI) and identify the temporal and spatial variation in groundwater pollution sources by using PCA and absolute principal component score-multiple linear regression (APCS-MLR) in the Hutuo River alluvial-pluvial fan region.

## 2. Materials and Methods

### 2.1. Study Area

The study area lies on the middle and upper part (37°50′–38°24′, 113°51′–114°55′) of the Hutuo River alluvial-pluvial fan in southwestern Hebei Province, China (Figure 1). Its total area covers approximately 2442 km^2^ and supports important agricultural and industrial activities. This region has a temperate semi-humid and semi-arid continental monsoon climate, with an average annual temperature of 13.3–15.0 °C [16], and the annual precipitation is mainly concentrated in June to September with precipitation of 450–750 mm (Appendix A).

The aquifer of the study region is part of the Hebei plain quaternary thick aquifer system, whose lithology is primarily composed of gravel, pebbles and coarse and fine sands [18]. The types of groundwater include fissure water and loose stratum pore water, and the groundwater flow runs from the northwest to the southeast. In the upper part of the Hutuo River alluvial-pluvial fan, the aquifer’s thickness is 10–35 m and its groundwater depth is 2–35 m, with very good hydraulic conductivity and water-richness. In the middle part of the Hutuo river alluvial-pluvial fan, the depth of the groundwater is greater, at 40–50 m, for which the main lithology comprises coarse sand and medium sand, and its water conductivity and water-richness are also good. The recharge sources of this groundwater include (1) atmospheric precipitation infiltration, (2) infiltration of river water, and (3) return of farmland irrigation water. The groundwater discharge is dominated by artificial pumping [17].

With the rapid urbanization and industrialization over recent years (according to the Shijiazhuang Statistical Yearbooks of 1995 and 2014, the land area in the city increased from 85 to 264 km^2^), and persistent groundwater over-exploitation has resulted in a decline of the regional groundwater table, such that large depression cones have formed locally [19]. Additionally, the chemical environment of shallow groundwater has been greatly changed by anthropogenic activities. This water chemistry has been transformed from HCO_3_-type water to HCO_3_-Cl, HCO_3_-SO_4_, and SO_4_-HCO_3_ types [18]. Several studies showed that the concentrations of groundwater NO_3_^−^, TH and total dissolved solids (TDS) in this region exceeded China’s drinking water standards [4,20,21]. Consequently, this is now a hidden hazard threatening the region’s economic and social development and the safety of drinking water for its local populations.

### 2.2. Groundwater Sampling and Laboratory Analyses

Groundwater samples were collected from the Hutuo River alluvial-pluvial fan region in January (i.e., dry season) of 2015, in October (i.e., transition season from rainy to dry) of 2015 and in August (i.e., rainy season) of 2016. Groundwater samples were obtained from three field surveys carried out in the dry, wet and transition seasons from the upstream to the downstream of Hutuo River, and consisted of 27 sampling sites (Figure 1). All the wells chosen for sampling groundwater are commonly used for domestic and/or agricultural purposes; their mean depth to the groundwater table was 9.78 m (min–max: 4.0–50.0 m). The samples’ pH and electrical conductivity (EC) were measured in the field using a WTW Multi 340i/SET multiparameter instrument (Germany).

Samples were collected by pumping groundwater from the wells, and stored in one 500 mL and one 1.5 L high-density polyethylene sampling bottle for analyzing the water quality parameters. The hydrochemical parameters of each groundwater sample were analyzed at the laboratory of Groundwater Mineral Water and Environmental Monitoring Center, based at the Institute of Hydrogeology and Environmental Geology, the Chinese Academy of Geological Sciences. The analyses of ions (i.e., nitrate [NO_3_^−^], nitrite [NO_2_^−^], chloride [Cl^−^], sulfate [SO_4_^2−^] and ammonia [NH_4_^+^]) were carried out using spectrophotometry (Perkin-Elmer Lambda 35, United States) and the detection limit values of the NO_3_^−^, NO_2_^−^, Cl^−^, SO_4_^2−^ and NH_4_^+^ were 0.664 mg/L, 0.003 mg/L, 1.0 mg/L, 0.75 mg/L and 0.026 mg/L, respectively. Analyses of cations (i.e., potassium [K^+^], sodium [Na^+^], calcium [Ca^2+^] and magnesium [Mg^2+^]) as well as two trace elements (Fe and Mn) in groundwater samples were performed using inductively coupled plasma-mass spectrometry (Agilent 7500ce ICP-MS, Tokyo, Japan) and the detection limit values of the K^+^, Na^+^, Ca^2+^, Mg^2+^, Fe and Mn were 0.05 mg/L, 0.01 mg/L, 4.0 mg/L, 3.0 mg/L, 0.0045 mg/L and 0.0005 mg/L, respectively. Bicarbonate [HCO_3_^−^] was determined using an acid–base titration and the detection limit value of the HCO_3_^−^ was 5.0 mg/L. Total dissolved solids (TDS) were quantified using gravimetric methods, and chemical oxygen demand (COD) was determined using alkaline permanganate oxidation, while the TH of each sample was measured according to the ethylene diamine tetraacetic acid titration method.

### 2.3. Data Analysis

#### 2.3.1. Principal Component Analysis (PCA)

PCA is a very powerful technique for variable reduction, dispensing with non-homogeneity in the sampling data, missing values and periodic trends, in order to elucidate spatial–temporal patterns in water quality and latent pollution sources [22]. A PCA works by extracting eigenvalues and their related loadings from the covariance matrix of the original variables to produce new orthogonal variables through varimax rotation, which are linear combinations of the original variables [23]. These new orthogonal variables (PC) allow data reduction with minimum loss of the original information, thereby providing information on the most meaningful parameters that describe the whole data set. Here, PCA was used to identify the main pollution sources of groundwater in different seasons.

#### 2.3.2. Absolute Principal Component Score-Multiple Linear Regression (APCS-MLR)

Absolute principal component score-multiple linear regression (APCS-MLR) can be used to estimate the contribution of each pollution source to the total, by combining MLR with the de-normalized APCS values produced by PCA and the measured concentrations of a particular pollutant [24]. After determining the number and identify of latent sources influencing the groundwater quality in the Hutuo River alluvial-pluvial fan (via the PCA), these source contributions were derived using the computations of APCS-MLR.

#### 2.3.3. Water Quality Index (WQI)

A robust WQI is an important indicator for assessing groundwater quality and its suitability for drinking purposes [14,25,26]. The WQI was calculated by assigning a weight (W*i*) to each water quality parameter according to its relative importance in the overall quality of groundwater for drinking purposes. Water quality standards mainly refer to the Grade III standard for groundwater quality in China [20]. When this standard was lacking for a given parameter, the World Health Organization (2011) [27] standards were used. The assigned weight (W*i*) and relative weight (RW*i*) for each parameter can be found in Appendix A.

The WQI was calculated as follows:(1)$$\mathrm{RW}i=\frac{Wi}{\sum_{i=1}^{n}Wi}$$
(2)$$Qi=\frac{Ci}{Si}\times 100$$
SI*i* = W*i* × Q*i*(3)
(4)WQI= ∑SIi
where Q*i* is the quality rating, C*i* is the concentration (mg/L) of each water quality parameter, S*i* is the water quality standard for each water quality parameter and SI*i* is the sub-index of the *i* parameter.

Calculated WQI values are usually classified into five categories (Appendix A): excellent, good, poor, very poor and unsuitable for human drinking [14].

## 3. Results and Discussion

### 3.1. Groundwater Quality Characteristic of the Hutuo River Alluvial-Pluvial Fan

Descriptive summary statistics of the groundwater quality data from the three sampling surveys that exceeded the national guidelines are given in Table 1. The mean groundwater pH and TDS values were 7.48 and 848.97 mg/L, respectively, among which 1.23% of pH and 22.22% of TDS samples exceeded the Grade III standard for groundwater quality in China [20]. Groundwater NO_3_^−^, SO_4_^2^^−^ and TH had mean values of 121.90 mg/L, 181.82 mg/L and 600.32 mg/L, respectively; 53.09% of NO_3_^−^, 18.52% of SO_4_^2^^−^ and 83.95% of TH samples surpassed the Grade III standard for groundwater quality in China [20]. According to this set of results, the mean concentrations of NO_3_^−^, SO_4_^2^^−^ and TH were very high in the Hutuo River alluvial-pluvial fan region. This indicated that its groundwater quality was generally impacted by human activity [6,13].

### 3.2. Water Quality Classification

In this study, the WQI was used to assess the groundwater quality according to China’s current quality standard [20]. Results for calculated WQIs during the dry, rainy and transition seasons are given in Table 2. The WQI ranged from 29.3 to 233.6. The maximum WQI value was obtained in the dry season and its minimum value was found in the rainy season.

In the dry season, there were 2, 15, 8 and 2 groundwater samples that had excellent, good, poor and very poor water quality, respectively, correspondingly accounting for 7.41%, 55.56%, 29.63% and 7.41% of the total groundwater samples. However, groundwater quality was improved in both the rainy and transition seasons, which may be attributed to the dilution caused by aquifer recharge in the two seasons [7]. In the rainy season, 2, 19, 4 and 2 of the groundwater samples were deemed of excellent, good, poor and very poor water quality, respectively; similarly, 3, 19 and 5 groundwater samples in the transition season were respectively classed as excellent, good and poor. It is worth noting that groundwater quality in the transition season (lacking any “very poor” samples) was slightly better than in the rainy season. A plausible explanation for this result is that rainwater infiltration of the aquifer lags behind the timing of rainfall; hence, in the transition season, the rainwater has completely entered the aquifer and diluted the groundwater pollutants’ concentrations.

### 3.3. Spatio-Temporal Variation in Groundwater Quality

According to the dry season’s WQI results, very poor water (sites 1 and 6) was considered to be dirty and could be defined as having a high pollution status (HP); poor water (sites 2, 4, 9–13 and 27) corresponded to moderate pollution (MP) and excellent and good water (i.e., at the other sites) was considered to be clean and thus defined as having a low pollution status (LP). Among the hydrochemical variables, those of pH, TDS, Cl^−^, TH, NO_3_^−^ and Fe were selected to analyze the spatial and temporal variation of groundwater quality in the study region.

Overall, groundwater quality varied spatially as follows: the high-polluted sites were mainly located in the upper part of the study area, the medium-polluted sites in both its upper and middle parts, and the low-polluted sites occurred only in its lower part (Figure 1). This spatial pattern may be related to the area’s groundwater depths. Previous studies have found that those areas with shallow groundwater depth were easily affected by human activities [4]. As shown in Figure 2, the concentration of TDS, Cl^−^, TH and NO_3_^−^ were the highest in the high-polluted sites, followed by the medium-polluted sites and then the low-polluted sites. By contrast, the pH and Fe levels were similar among the three pollution types.

Trends in the temporal variation of groundwater quality were inconsistent in the three pollution types (Figure 2), in that Cl^−^, TH and NO_3_^−^ concentrations in the high- and medium-polluted sites were higher in the dry than the rainy season. This finding is probably also related to the dilution effect of groundwater pollutants by rainwater [7]; however, no regularity was apparent at the low-pollution sites. The pH at the sites of all three pollution types showed that the transition season was higher than the dry and rainy season. The concentrations of TDS and Fe did not show significant regularity for the three types of polluted sites.

### 3.4. Correlations between the Water Quality Variables

Figure 3a–c shows the correlation matrixes of the 14 variables in each of the three seasons. A significant negative correlation (−0.40) between pH and HCO_3_^−^ occurred in the dry season, and also between pH and TDS, TH, Ca^2+^, Cl^−^, SO_4_^2−^, NO_3_^−^ and HCO_3_^−^ (r-values ranging from −0.39 to −0.78) in both the rainy season and transition season. The stronger correlation between pH and other indicators found for the rainy and transition seasons are mainly due to the massive amount of rainwater that permeates into the aquifer, as this would have promoted water–rock interactions, raising the concentration of ions and lowering the pH of the groundwater.

Positive and relatively strong correlations were found among TDS, K^+^, Na^+^, Ca^2+^, Mg^2+^, Cl^−^, SO_4_^2−^, NO_3_^−^, HCO_3_^−^ and TH in the dry season (r-values: 0.54–0.97), rainy season (r-values: 0.61–0.96), and transition season (r-values: 0.67–0.95). These high correlations implied that groundwater chemistry was mainly controlled by these ions that came from common sources. Cl^−^, SO_4_^2−^ and NO_3_^−^ are important indicators of the impact of human activities upon groundwater quality [28]. Therefore, the groundwater quality in the study region has clearly been affected by human activities. 

The presence of Fe in groundwater possibly represents metal pollution derived from industrial effluents [29]. Positive and relatively strong correlations were found between Fe and Mn (r-values: 0.65–0.88) in the three seasons, which indicates they shared a common source of pollution.

### 3.5. Identifying the Main Groundwater Pollution Sources via PCA

PCA with varimax rotation explained 79.69%, 83.12% and 79.68% of the total variance in the dry season, transition season and rainy season, respectively (Table 3). For the dry season, factor I explains 57.80% of the total variance and includes a strong positive loading of TDS, TH, NO_3_^−^, Ca^2+^ and Cl^−^ and a moderate positive loading of K^+^, SO_4_^2−^, Na^+^, Mg^2+^ and COD.

Nitrate contamination in groundwater is attributed to anthropogenic activities including municipal wastewater discharge, agricultural runoff with a significant amount of chemical fertilizer and atmospheric deposition [30,31]. In this study region, the higher NO_3_^−^ concentration (the mean value is 121.90 mg/L) in groundwater possibly comes mainly from domestic wastewater [17]. Indeed, during the study period, we found that the villages and towns in the upstream of the Hutuo River area did not construct a network of sewage pipes, and large volumes of domestic sewage were drained directly into nearby rivers and ditches without treatment; this domestic sewage inevitably infiltrates into groundwater. In addition, in the dry season, agricultural chemical fertilizers are not a main pollution source of nitrate in groundwater due to the lack of the driving force of rainfall runoff. Therefore, it is difficult for fertilizer to seep into the groundwater.

Chloride may originate from domestic sewage, chemical fertilizers, manure, road salt and the natural dissolution of minerals [32,33]. In the study area, the chemical fertilizers and road salt were possibly not the main pollution sources of Cl^−^. In this region, chlorine fertilizer was rarely applied and road salts was primary used in urban areas; after the snow melts on the road, it flows directly into the urban sewage treatment plant through the city’s sewage network. Thus, the two do not have a direct impact on groundwater in the Hutuo River basin. Therefore, Cl^−^ most likely comes from domestic wastewater in the region.

In addition, the higher concentration of TH, Ca^2+^, K^+^, SO_4_^2−^, Na^+^ and Mg^2+^ may be related to the a large discharge of domestic sewage [33]. Due to the untreated discharge of domestic sewage, a large amount of K^+^ and Na^+^ ions enters the soil layer along with the liquid medium, thereby displacing Ca^2+^ and Mg^2+^ ions into the groundwater [19]. Accompanied by cationic exchange, the ions of K^+^ and Na^+^ are adsorbed onto the aquifer and the ions of Ca^2+^ and Mg^2+^ are desorbed into groundwater. Thus, the domestic sewage input to the aquifer leads to an increase in Ca^2+^, Mg^2+^ and TH in the groundwater. Therefore, PC1 is considered to denote to the domestic sewage.

Factor 2, which describes 11.33% of the total variance, has a high positive loading for Fe and Mn. Fe and Mn could be a response to a dominant reduction in the conditions in the aquifer [13]. In addition, the higher concentration of Fe and Mn in the water environment may originate from industrial effluents [34]. Indeed, there are steel processing plants in the upper areas of the region (located in Pingshan county). Many samples of groundwater in the upper areas of the region such as G1 (1.165 mg/L), G3 (0.577 mg/L), G6 (2.308 mg/L) and G7 (0.369 mg/L) had high concentrations of Fe in the dry season (Figure 1). In this study area, the aquifer system had an oxidizing environment (the average concentration of dissolved oxygen in groundwater in the three season was 6.82 mg/L). Therefore, PC2 refers to industrial sewage pollution.

Factor 3, which describes 10.56% of the total variance, has a strong positive loading for HCO_3_^−^ and a strong negative loading for pH, and weak positive loadings on Mg^2+^, Ca^2+^ and TH. In these regions, the higher concentration of HCO_3_^−^, Mg^2+^ and Ca^2+^ in groundwater is in relation with cation exchange and water–rock interaction [1,13]. The intensive cation exchange occurring in the Hutuo River area influences groundwater HCO_3_^−^, Mg^2+^ and Ca^2+^ concentration. In recent years, the groundwater table has dropped rapidly along with the rapid development of industry and agriculture, added to the thickness of the vadose zone and enhanced the cation exchange process. This causes an increased in the concentration of HCO_3_^−^, Mg^2+^ and Ca^2+^ in groundwater. Therefore, PC3 is considered to denote water–rock interactions.

For the wet season, factor I explains 65.31% of the total variance and includes a strong negative loading of pH and a strong positive loading of Ca^2+^, NO_3_^−^ and TH, and a moderate positive loading of HCO_3_^−^, Mg^2+^, TDS, Cl^−^ and SO_4_^2−^, likely representing the domestic sewage and water–rock interactions. Factor 2, which describes 10.00% of the total variance, has a high positive loading for COD and K^+^ and a moderate positive loading for TDS, Na^+^, Cl^−^, SO_4_^2−^, Ca^2+^ and TH. The higher concentration of COD in groundwater is in relation to the non-point pollution. Young et al. (2011) [35] found that the COD is dominant pollutants from the urban highways. Zhang et al. (2013) [36] also found that the concentrations of COD on asphalt roads were 3–5 times higher than on concrete roads. In the research region, road materials in urban and rural areas are mainly asphalt. Thus, we assume that the higher concentration of COD in the groundwater may be the result of nonpoint pollution. In addition, the higher concentrations of K^+^, TDS, Na^+^, Cl^−^, SO_4_^2−^, Ca^2+^ and TH may be related to nonpoint pollution. Rainfall runoff carries a large amount of ions that were deposited on the road and leaches into the groundwater, and the ion exchange ability was also increases during the runoff infiltration process, which in turn increases the concentration of these ions. Therefore, PC2 is considered to denote “nonpoint pollution sources”. Factor 3, which describes 7.80% of the total variance, has a high positive loading for Mn and Fe, likely representing industrial sewage pollution.

The pollution pattern in the transition season was similar to that in the wet season. Factor I explains 54.64% of the total variance and includes a strong positive loading of TH, HCO_3_^−^, Ca^2+^, NO_3_^−^, TDS, SO_4_^2−^ and Mg^2+^, a strong negative loading of pH and a moderate positive loading of Na^+^ and Cl^−^, likely representing domestic sewage and water–rock interactions. Factor 2, which describes 13.32% of the total variance, has a high positive loading for COD and K^+^ and has a moderate positive loading for Cl^−^, Na^+^ and TDS. Thus, PC2 is considered to denote “nonpoint pollution sources”. Factor 3, which describes 11.72% of the total variance, has a high positive loading for Mn and Fe, likely representing industrial sewage pollution.

### 3.6. Source Apportionment Using APCS-MLR

After identifying the pollution sources, the contributions of each source to each water quality variable were calculated using APCS-MLR. As shown in Table 4, most water quality parameters in the dry season were mainly influenced by domestic sewage (60.84% of TDS, 34.33% of K^+^, 57.98% of Na^+^, 50.62% of Ca^2+^, 41.66% of Mg^2+^, 54.09% of Cl^−^, 55.20% of NO_3_^−^, 41.32% of SO_4_^2−^, 41.40% of TH and 71.06% of COD), industrial sewage (36.12% of Mn and 55.59% of Fe) and water–rock interactions (31.08% of pH and 43.64% of HCO_3_^−^).

For the wet season, most water quality variables were primarily influenced by domestic sewage and water–rock interactions (26.72% of pH, 68.15% of Ca^2+^, 79.73% of Mg^2+^, 59.72% of NO_3_^−^, 37.98% of HCO_3_^−^ and 78.85% of TH), nonpoint pollution (60.02% of TDS, 53.31% of K^+^, 57.36% of Na^+^, 44.50% of Cl^−^, 42.92% of SO_4_^2−^ and 43.05% of COD) and industrial sewage (53.73% of Mn and 52.90% of Fe).

For the transition season, most water quality variables were primarily influenced by domestic sewage and water–rock interactions (61.11% of pH, 46.97% of TDS, 63.19% of Ca^2+^, 64.54% of Mg^2+^, 55.57% of NO_3_^−^, 62.51% of SO_4_^2−^, 41.16% of HCO_3_^−^ and 41.72% of TH), nonpoint pollution (32.60% of K^+^, 55.00% of Na^+^, 45.62% of Cl^−^ and 57.09% of COD) and industrial sewage (38.71% of Mn and 39.56% of Fe).

Based on all of the above discussion, the major source was domestic sewage in the three seasons in the Hutuo River alluvial-pluvial fan area. As shown in Table 4, the contribution of unidentified sources to pollution in the Hutuo River alluvial-pluvial fan area for all water quality variables ranged from 0.41% to 68.92% for the dry season, from 9.47% to 73.28% for the wet season, and from 11.88% to 61.29% for the transition season. The contribution ratio of the unidentified sources to groundwater quality parameters was higher than the river [27,37,38], which may be due to the fact that the sources of groundwater pollution are varied and complex, and that the statistical technology has certain limitations. Therefore, field surveys are important to further identify the pollution sources.

## 4. Conclusions

In this study, the water quality index and multivariate statistical techniques (PCA and APCS-MLR) were applied to assess the spatial and temporal variation in groundwater quality and to trace the pollution sources in different seasons in the Hutuo River alluvial-pluvial fan, using three field sampling data sets. Results showed that the mean concentration of groundwater NO_3_^−^, SO_4_^2^^−^ and TH were 121.90 mg/L, 181.82 mg/L and 600.32 mg/L, respectively; 53.09% of NO_3_^−^, 18.52% of SO_4_^2^^−^ and 83.95% of the TH samples exceeded the Grade III standard for groundwater quality in China (GB/T 14848-2017). This indicated that the groundwater quality was seriously affected by human activity in the Hutuo River alluvial-pluvial fan region.

We used WQI to assess groundwater quality. The results showed that the calculated WQI ranged from 29.3 to 233.6 and that the maximum value of the WQI was in the dry season and the minimum value was in the rainy season. The groundwater quality in the transition season and rainy season was better than in the dry season due to the rain having diluted the pollutants in the groundwater.

The spatial variations of groundwater quality showed that the concentration of TDS, Cl^−^, TH and NO_3_^−^ are the highest in the high-polluted sites, followed by the medium-polluted sites and the low-polluted sites; however, the pH and Fe concentration shows no significant changes between the three pollution types. The temporal variations of groundwater quality were controlled by the rainwater dilution and showed that the temporal variation patterns’ parameters were not consistent in different season.

PCA results show that the primary sources of groundwater pollution were domestic sewage, industrial sewage and water–rock interactions in the dry season. However, in the rainy and transition seasons, nonpoint pollution has an important effect on groundwater quality, and the main pollution sources changed to domestic sewage and water–rock interactions, as well as nonpoint pollution and industrial sewage. Based on the APCS-MLR, we found that most water quality parameters were primarily influenced by domestic sewage. Therefore, in order to prevent the continuous deterioration of groundwater quality it is necessary to control the discharge of domestic sewage in the Hutuo River alluvial-pluvial fan region.

## Figures and Tables

**Figure 1 ijerph-17-01055-f001:**
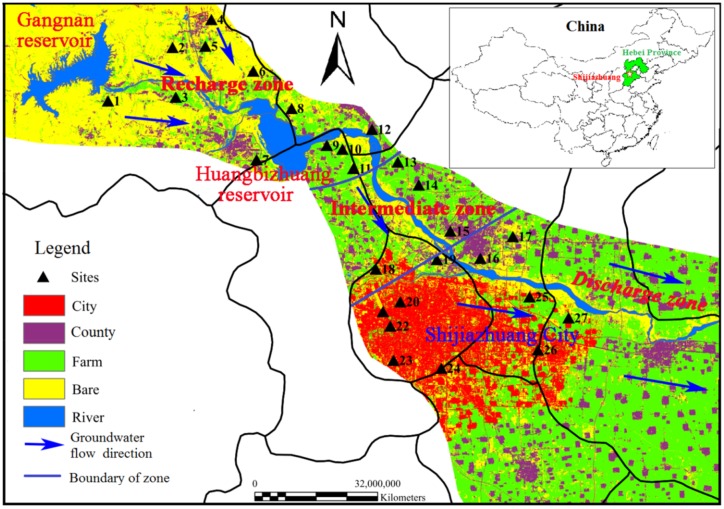
Water quality monitor sites and land use types of the Hutuo River alluvial-pluvial fan in China.

**Figure 2 ijerph-17-01055-f002:**
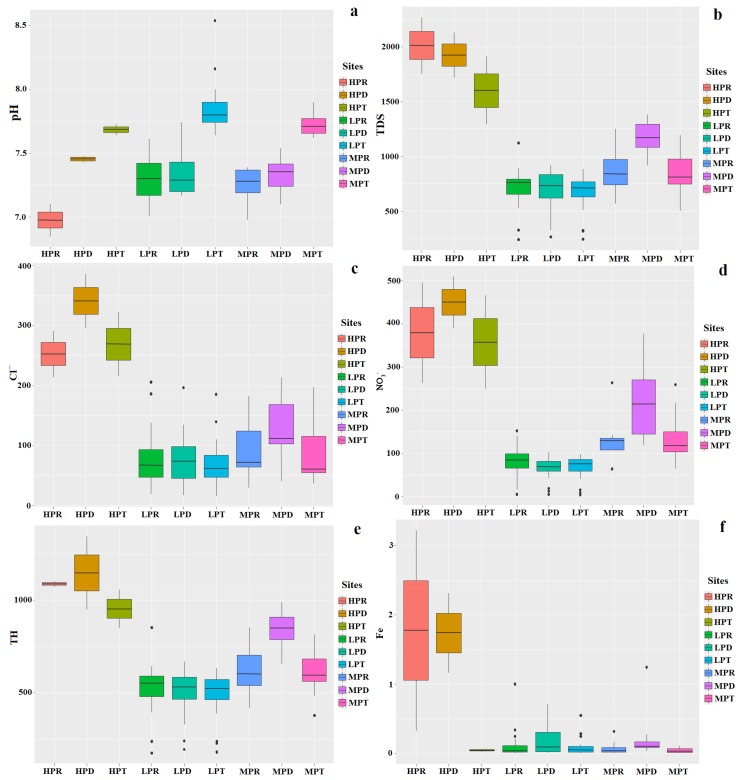
Spatial and temporal pattern of the (**a**) pH, (**b**) TDS, (**c**) Cl^−^, (**d**) NO_3_^−^, (**e**) TH and (**f**) Fe values of r in the Hutuo River alluvial-pluvial fan. Note: HPR = High-polluted sites in rainy season; HPD = High-polluted sites in dry season; HPT = High-polluted sites in transition season; LPR = Low-polluted sites in rainy season; LPD = Low-polluted sites in dry season; LPT = Low-polluted sites in transition season; MPR = Medium-polluted sites in rainy season; MPD = Medium-polluted sites in dry season; MPT = Medium-polluted sites in transition season.

**Figure 3 ijerph-17-01055-f003:**
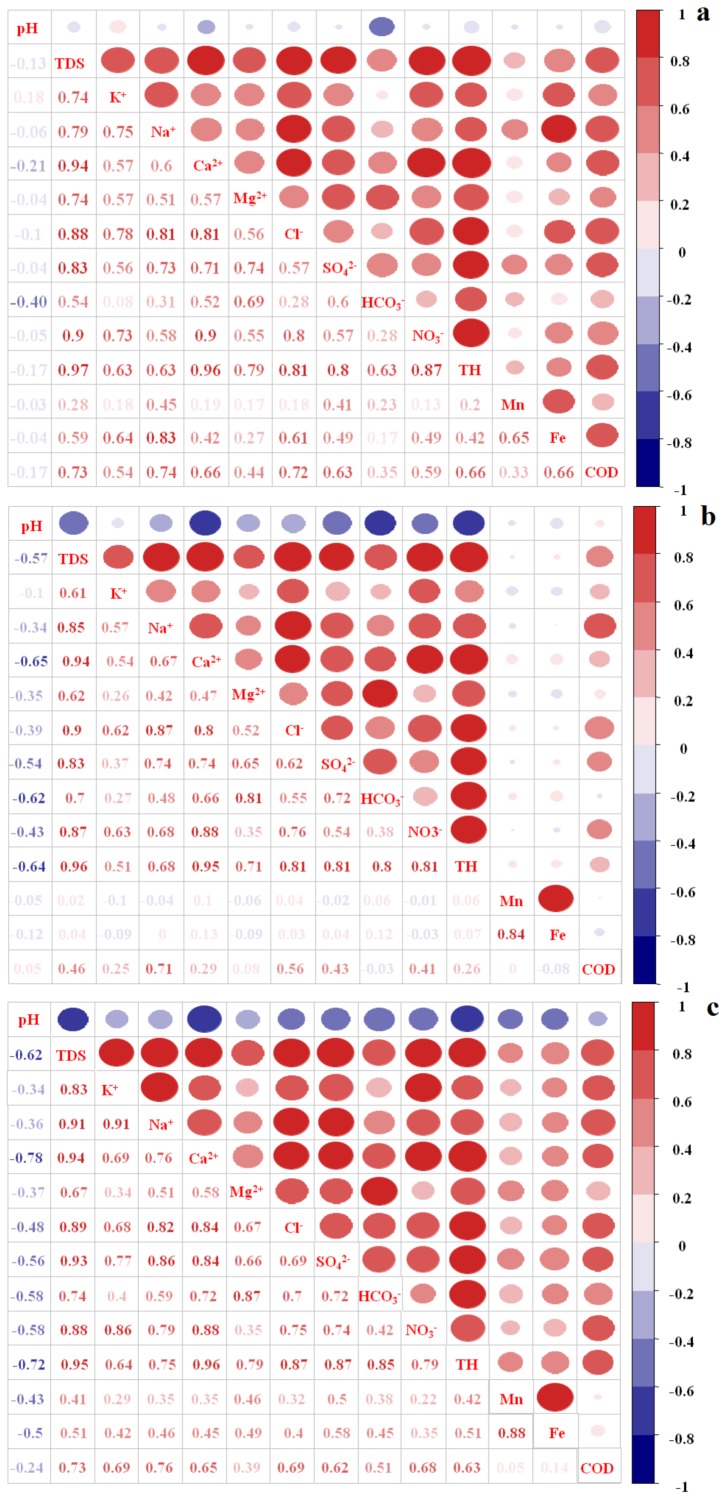
Pearson’s correlation matrix of physicochemical and hydrochemical parameters of groundwater samples in the dry season (**a**), transition season (**b**) and rainy season (**c**).

**Table 1 ijerph-17-01055-t001:** Groundwater quality parameters and summary basic statistics of the Hutuo River alluvial-pluvial fan.

Parameters	Mean	S.D.	Min	Max	Standard	Below Standards for All Sites (%)	Units
pH	7.48	0.30	6.85	8.54	6.5–8.5	1.23	
EC	1303.32	559.78	370.00	3530.00	-	-	μs/cm
Na^+^	46.40	43.59	8.88	262.40	200	1.23	mg/L
Ca^2+^	175.69	66.08	51.57	359.80	-	-	mg/L
Mg^2+^	39.25	18.53	10.23	108.40	-	-	mg/L
Cl^−^	100.59	73.67	15.85	385.90	250	4.94	mg/L
SO_4_^2−^	181.82	93.17	21.77	530.80	250	18.52	mg/L
HCO_3_^−^	320.69	71.09	153.30	462.10	-	-	mg/L
NO_3_^−^	121.90	105.92	5.04	509.00	88.6	53.09	mg/L
NO_2_^−^	0.019	0.103	0.002	0.920	3.29	0	mg/L
TH	600.32	217.17	178.10	1345.00	450	83.95	mg/L
TDS	848.97	381.21	239.10	2269.00	1000	22.22	mg/L
COD	0.922	0.378	0.320	2.240	3.0	0	mg/L
Fe	0.216	0.478	0.010	3.216	0.3	17.28	mg/L
Mn	0.008	0.016	0.001	0.120	0.1	1.23	mg/L

Note: Mean: average value; S.D: standard deviation; Min: minimum value; Max: maximum; Standard is grade III standard for groundwater quality in China (GB/T 14848-2017). EC: electrical conductivity; TH: total hardness; TDS: total dissolved solids; COD: chemical oxygen demand.

**Table 2 ijerph-17-01055-t002:** Statistics of water quality index classification for different seasons in the Hutuo River alluvial-pluvial fan.

WQI Range	Dry Season		Rainy Season		Transition Season	
	Number	Rate (%)	Number	Rate (%)	Number	Rate (%)
Excellent water	2	7.41	2	7.41	3	11.11
Good water	15	55.56	19	70.37	19	70.37
Poor water	8	29.63	4	14.81	5	18.52
Very poor water	2	7.41	2	7.41	0	0.00
Water unsuitable for drinking purposes	0	0.00	0	0.00	0	0.00
Sum	27		27		27	

Note: WQI: water quality index.

**Table 3 ijerph-17-01055-t003:** Loadings of 14 selected variables on VARIMAX rotated factors of the three seasons.

Parameters	Dry Season			Wet Season			Transition Season		
	PC1	PC2	PC3	PC1	PC2	PC3	PC1	PC2	PC3
pH	0.041	−0.121	−0.745	−0.879	−0.021	−0.191	−0.810	0.133	−0.001
TDS	0.952	0.221	0.185	0.623	0.703	0.296	0.799	0.576	0.057
K^+^	0.748	0.285	−0.402	0.099	0.779	0.353	0.136	0.777	0.101
Na^+^	0.721	0.525	−0.071	0.376	0.694	0.484	0.608	0.636	0.037
Ca^2+^	0.891	0.087	0.359	0.811	0.520	0.179	0.815	0.403	0.167
Mg^2+^	0.709	0.078	0.380	0.604	0.323	0.413	0.756	0.107	−0.064
Cl^−^	0.883	0.189	−0.028	0.548	0.672	0.232	0.599	0.697	0.040
NO_3_^−^	0.906	0.055	−0.008	0.793	0.347	0.079	0.814	0.194	0.189
SO_4_^2−^	0.735	0.356	0.263	0.530	0.619	0.406	0.768	0.413	−0.040
HCO_3_^−^	0.402	0.107	0.798	0.638	0.352	0.490	0.902	0.030	0.098
TH	0.923	0.093	0.330	0.789	0.505	0.316	0.903	0.356	0.111
COD	0.694	0.423	0.054	0.259	0.888	−0.049	−0.021	0.866	−0.096
Mn	0.056	0.916	0.070	0.104	0.077	0.921	−0.036	0.018	0.934
Fe	0.297	0.820	0.210	0.251	0.246	0.834	0.180	0.012	0.932
Eigenvalue	8.09	1.59	1.48	9.14	1.40	1.09	7.65	1.87	1.64
% Total variance	57.80	11.33	10.56	65.31	10.00	7.80	54.64	13.32	11.72
Cumulative % variance	57.80	69.13	79.69	65.31	75.31	83.12	54.64	67.96	79.68

**Table 4 ijerph-17-01055-t004:** Source contribution (in %) of each variable in three seasons in the Hutuo River alluvial-pluvial fan.

Parameters	Potential Pollution Source in the Dry Season (^a^)				R^2^	Potential Pollution Source in the Wet Season (^b^)				R^2^	Potential Pollution Source in the Transition Season (^c^)				R^2^
	S1	S2	S3	US ^d^		S1	S2	S3	US		S1	S2	S3	US	
pH	0.00	0.00	31.08	68.92	0.550	26.72	0.00	0.00	73.28	0.407	61.11	0.00	0.00	38.89	0.681
TDS	60.84	4.31	34.44	0.41	0.991	14.08	60.02	8.05	17.85	0.763	46.97	1.69	39.46	11.88	0.979
K^+^	34.33	3.26	21.81	40.60	0.830	16.00	53.31	0.00	30.69	0.600	18.41	32.60	0.00	48.99	0.532
Na^+^	57.98	15.39	0.00	26.62	0.857	20.24	57.36	0.00	22.40	0.765	24.01	55.00	0.00	20.99	0.693
Ca^2+^	50.62	0.00	38.85	10.53	0.865	68.15	12.05	10.33	9.47	0.634	63.19	2.22	0.00	34.59	0.841
Mg^2+^	41.66	0.00	33.21	25.13	0.651	79.73	0.00	0.00	20.27	0.482	64.54	0.00	0.00	35.46	0.537
Cl^−^	54.09	4.06	0.00	41.85	0.833	32.74	44.50	0.00	22.76	0.590	21.99	45.62	1.43	30.96	0.815
NO_3_^−^	55.20	0.00	0.00	44.80	0.827	59.72	9.63	0.00	30.65	0.598	55.57	0.00	0.00	44.43	0.654
SO_4_^2−^	41.32	4.57	28.27	25.84	0.732	24.16	42.92	10.04	22.88	0.810	62.51	0.00	0.00	37.49	0.679
HCO_3_^−^	7.12	0.00	43.64	49.24	0.824	37.98	9.95	0.00	52.06	0.489	41.16	0.00	15.56	43.28	0.672
TH	41.40	0.96	39.99	17.66	0.974	78.85	10.69	0.00	10.47	0.626	41.72	39.49	0.00	18.79	0.935
COD	71.06	13.90	0.00	15.04	0.645	0.00	43.05	0.00	56.95	0.523	13.57	57.09	0.00	29.34	0.381
Mn	0.00	36.12	13.54	50.34	0.800	0.00	0.00	53.73	46.27	0.411	0.00	0.00	38.71	61.29	0.556
Fe	33.64	55.59	0.00	10.77	0.888	0.00	0.00	52.90	47.10	0.503	0.00	0.00	39.56	60.44	0.605

Note: (^a^) S1 = domestic wastewater, S2 = industrial sewage; S3 = water–rock interactions; (^b^) S1 = domestic sewage and water–rock interactions; S2 = nonpoint pollution; S3 = industrial sewage; (^c^) S1 = domestic sewage and water–rock interactions; S2 = nonpoint pollution; S3 = industrial sewage; (^d^) US = unidentified sources.

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
