# Peer review of "Spatio-Temporal Variation of Groundwater Quality and Source Apportionment Using Multivariate Statistical Techniques for the Hutuo River Alluvial-Pluvial Fan, China"

_ijerph, 2020, doi:10.3390/ijerph17031055_

Round 1

Reviewer 1 Report

General comments:

The works present an interesting analysis of a set of chemical data collected in three different seasons of a year. The introduction points out the necessity of identify the sources of pollution of the GW in the area in order to be able to limit and/or control them. The authors propose PCA methods and define the objectives of the work. In my opinion the introduction is correct but it would be nice to extend and elaborate it more. There are several grammar mistake along the document, just some extra words in some cases or a wrong order in other cases, the authors should revise the document and correct them.

It would be desirable to include hydrogeological information (heads, piezometric information, recharge and discharge areas…) in some figure in order to provide the general idea of the groundwater flow.

There are editing errors along the document, many of them are missing spaces, or grammar mistakes, but it is needed to devote some extra time to the manuscript in order to remove these mistakes.

The caption of the figures is not clear enough. The multi-figures should be named and explained in the caption individually.

The cites are scarce. In many occasions the authors refer to a particular study cases to support general statements. Frequently the cite used to support a statement is not a specific study about the statement itself. This has to be improve carefully.

The site, sampling, and data analysis are presented in a clear way. PCA is useful to elucidate spatio-temporal patterns but, one year of observations are not enough for conclude temporal pattern.

I suggest the authors to evaluate the possibility of move part of the information (the paragraph describing the consequences of the over-exploitation) to the introduction for describe the context and highlights the relevance of the study. 

Table 1 summarizes perfectly the data set; I think that the repetition of the information in the text is a little redundant. I suggest to limit the information in the text to the parameters that authors want to emphasize (NO3-, SO4-2, and TH).

The authors declare that the presence of Fe and Mn in groundwater is due to pollution from industrial effluents. Nevertheless, Fe and Mn are very frequently detected in anoxic and/or anaerobic groundwater. It should be interesting to have the oxygen values of the samples.

In section 3.4, authors point out the strong correlation between Ca+2, Mg+2, Cl-, SO4-2, HCO3- and TH. Taking into account the total hardness definition it could not be differently.

The cationic exchange explained in the paragraph from line 267 to line should be elaborate and it would be appropriate to provide references and data supporting it. The authors need to explain this and add references supporting it.

Fe and Mn could be a response of a dominant reducing conditions in the aquifer. In the wet season the natural recharge would provide oxic water and oxidize Fe(II) to Fe (III) and Mn (II) to Mn (IV) and, therefore, reduce Fe and Mn dissolved concentrations. This could also explain the distribution of this two metals. How do the authors know that Fe and Mn presence is due to the industrial effluents?

Concluding that temporal variations are controlled by a given parameter with just one year observations is a strong assumption. It would be much more appropriate to state a hypothesis based on the one year observations. Stablish a funded temporal pattern would require further work and more data along several years.

Authors contributions and Funding have not been modified from the template. You need to fill these sections.

Specifics Comments:

Line 21: primary sources of groundwater were…, Do authors mean “primary sources of groundwater pollution were…”?.

Line 36: a space is missing before the two square brackets.

Line 40:  When authors summarize the natural processes affecting groundwater quality they cite just one local work; I would appreciate adding some more cites from other regions.

Line 41: “According to Quin et al. (2013), the major factors controlling groundwater quality were Holocene transgression and…” while the sentence is completely general the cited work focuses in costal alluvial aquifers of the lower Liaohe River Plain. I suggest to specify the area for which this factors are determinants or to add more cites in order to generalize. The number of the cite is missing in the sentence.

Line 43: The work cited is focused in a particular region, please specify it instead of formulate a very general affirmation based in a particular observation.

Line 46: “..the major factors affecting it and along with the corresponding sources of pollution” I think “along with” can be removed from the sentence.

Line 51: a space is missing before the square brackets

Line 60: The total area of the alluvial…

Line 63: It is always preferable to cite the information source. The annual temperature of the region is not the objective of the work you are citing even though it mentioned it. The source of the evaporation data is missing.

Line 68:  The cited work for describe the aquifer is a recent study about nitrate in the area, is not the proper cite. You can see “Quaternary Aquifer of the North China Plain—assessing and achieving groundwater resource sustainability” from Foster et al. 2004, and works therein.

Line 69: a space is missing before the square brackets.

Line 70-line 78: Cites are missing. What is water-richness?

Line 79-Line 87: Cites are missing.

Line 124: A space is missing before the square brackets.

Line 129: A space is missing before the square brackets.

Line 132: A space is missing before the square brackets, and in the next sentence is not in the correct order.

Line 149: I would divide the sentence in two

Line 150: A space is missing before the square brackets.

Line 171: Correct the sentence

Line 239: A space is missing before the square brackets.

Line 247: A space is missing before the brackets.

Line 252: A space is missing before the brackets.

Line 256: Correct the sentence

Line 260: Add additional references for the origin of Cl- in groundwater, and separate the cites from the las word.

Line 261: Rewrite the sentence, it is grammatically wrong.

Line 162: Rewrite the sentence, you use “mainly” three times.

Line 268: What do you mean with “unscientific discharge”?

Line 276: A space is missing before the square brackets and another space is missing before the brackets.

Line 278: Spaces are missing after G3, G6, and season.

Line 316: A space is missing before the brackets

Line 317, 318 y 319: spaces are missing after the after the commas and before brackets.

Line 341: A space is missing before the brackets

Figure 1: It would be great if the river was included in the figure 1-1. The caption of the figure is incomplete, name figure 1A and 1B and specify in the caption what is display in each figure. It would be very clarifying to have the piezometric head of the area or at least the main recharge areas and the general groundwater flow direction.

Figure 2: A name or signed to identify each figure is missing. The titles of the graphics and the number in the axis are too small. The legend is the same for the 6 figures, I suggest to make one much bigger to avoid duplicate the information. The caption is heavily incomplete; it should provide information about what is being shown in each figure.

Supplementary Table 2: the values and categories of WQI can be added to table 2 (just adding another column.

Author Response

Reviewer #1:

General comments:

The works present an interesting analysis of a set of chemical data collected in three different seasons of a year. The introduction points out the necessity of identify the sources of pollution of the GW in the area in order to be able to limit and/or control them. The authors propose PCA methods and define the objectives of the work. In my opinion the introduction is correct but it would be nice to extend and elaborate it more. There are several grammar mistake along the document, just some extra words in some cases or a wrong order in other cases, the authors should revise the document and correct them.

It would be desirable to include hydrogeological information (heads, piezometric information, recharge and discharge areas…) in some figure in order to provide the general idea of the groundwater flow.

There are editing errors along the document, many of them are missing spaces, or grammar mistakes, but it is needed to devote some extra time to the manuscript in order to remove these mistakes.

The caption of the figures is not clear enough. The multi-figures should be named and explained in the caption individually.

Response: We have named each figures and explained in the caption individually

The cites are scarce. In many occasions the authors refer to a particular study cases to support general statements. Frequently the cite used to support a statement is not a specific study about the statement itself. This has to be improve carefully.

Response: We have added the References.

The site, sampling, and data analysis are presented in a clear way. PCA is useful to elucidate spatio-temporal patterns but, one year of observations are not enough for conclude temporal pattern.

Response: Thanks for your suggestion. In the paper, we mainly discuss the season variation of groundwater quality in Hutuo River region.

I suggest the authors to evaluate the possibility of move part of the information (the paragraph describing the consequences of the over-exploitation) to the introduction for describe the context and highlights the relevance of the study. 

Table 1 summarizes perfectly the data set; I think that the repetition of the information in the text is a little redundant. I suggest to limit the information in the text to the parameters that authors want to emphasize (NO3-, SO4-2, and TH).

Response: We have been removed some parameter descriptions in 3.1 section.

The authors declare that the presence of Fe and Mn in groundwater is due to pollution from industrial effluents. Nevertheless, Fe and Mn are very frequently detected in anoxic and/or anaerobic groundwater. It should be interesting to have the oxygen values of the samples.

Fe and Mn could be a response of a dominant reducing conditions in the aquifer. In the wet season the natural recharge would provide oxic water and oxidize Fe(II) to Fe (III) and Mn (II) to Mn (IV) and, therefore, reduce Fe and Mn dissolved concentrations. This could also explain the distribution of this two metals. How do the authors know that Fe and Mn presence is due to the industrial effluents?

Response: In this study area, the aquifer system had an oxidizing environment (the average DO in groundwater in three season was 6.82 mg/L) and Fe and Mn could be a response of a dominant reducing conditions in the aquifer.In addition, there are a steel processing plants in the upper areas of the region. and some samples of groundwater in the upper areas of the region such as G1 (1.165mg/L), G3 (0.577mg/L), G6 (2.308mg/L) and G7 (0.369mg/L) with high concentrations of Fe. Therefore, we infer that the Fe  and Mn was from industrial sewage pollution.

In section 3.4, authors point out the strong correlation between Ca+2, Mg+2, Cl-, SO4-2, HCO3-and TH. Taking into account the total hardness definition it could not be differently.

The cationic exchange explained in the paragraph from line 267 to line should be elaborate and it would be appropriate to provide references and data supporting it. The authors need to explain this and add references supporting it.

Response: We have added the reference.  

Concluding that temporal variations are controlled by a given parameter with just one year observations is a strong assumption. It would be much more appropriate to state a hypothesis based on the one year observations. Stablish a funded temporal pattern would require further work and more data along several years.

Response: Thanks for your suggestion. In the paper, we mainly discuss the season variation of groundwater quality in Hutuo River region.

Authors contributions and Funding have not been modified from the template. You need to fill these sections.

Response: We have added the Authors contributions and Funding.

Specifics Comments:

Line 21: primary sources of groundwater were…, Do authors mean “primary sources of groundwater pollution were…”?.

Response: We have revised " primary sources of groundwater were " as " primary sources of groundwater pollution were " in line 25.

Line 36: a space is missing before the two square brackets.

Response: We have added a space.

Line 40:  When authors summarize the natural processes affecting groundwater quality they cite just one local work; I would appreciate adding some more cites from other regions.

Response: Thanks for your suggestion. We have revised " Qin et al. (2013), the major factors controlling groundwater quality were holocene transgression and mixing, surface water infiltration, multi-factor processes, rainfall regimes and agricultural fertilizer contamination, and geogenic fluoride enrichment. " as " According to Qin et al. [1] the major factors controlling groundwater quality in a coastal alluvial aquifers of the lower Liaohe River Plain, China were holocene transgression and mixing, surface water infiltration, multi-factor processes, rainfall regimes and agricultural fertilizer contamination, and geogenic fluoride enrichment." in line 45-48.

Line 41: “According to Quin et al. (2013), the major factors controlling groundwater quality were Holocene transgression and…” while the sentence is completely general the cited work focuses in costal alluvial aquifers of the lower Liaohe River Plain. I suggest to specify the area for which this factors are determinants or to add more cites in order to generalize. The number of the cite is missing in the sentence.

Response: Thanks for your suggestion. We have revised " Qin et al. (2013), the major factors controlling groundwater quality were holocene transgression and mixing, surface water infiltration, multi-factor processes, rainfall regimes and agricultural fertilizer contamination, and geogenic fluoride enrichment. " as " According to Qin et al. [1] the major factors controlling groundwater quality in a coastal alluvial aquifers of the lower Liaohe River Plain, China were holocene transgression and mixing, surface water infiltration, multi-factor processes, rainfall regimes and agricultural fertilizer contamination, and geogenic fluoride enrichment." in line 45-48.

Line 43: The work cited is focused in a particular region, please specify it instead of formulate a very general affirmation based in a particular observation.

Response: We have revised " Herojeet et al. [7] recently showed that groundwater chemistry was strongly controlled by water rock interaction, ion exchange and leaching of parent materials, agricultural runoff, and the seepage of industrial and domestic wastes material. " as " Herojeet et al. [7] recently showed that groundwater chemistry in an alluvial aquifer of Nalagarh Valley , India was strongly controlled by water rock interaction, ion exchange and leaching of parent materials, agricultural runoff, and the seepage of industrial and domestic wastes material." in line 48-51.

Line 46: “..the major factors affecting it and along with the corresponding sources of pollution” I think “along with” can be removed from the sentence.

Response: We have revised " An effective way to mitigate and control the continuous deterioration of groundwater quality is to identify the major factors affecting it and along with the corresponding sources of pollution." as" An effective way to mitigate and control the continuous deterioration of groundwater quality is to identify the major influencing factors and the pollution sources." in line 52-54.

Line 51: a space is missing before the square brackets

Response: We have added a space.

Line 63: It is always preferable to cite the information source. The annual temperature of the region is not the objective of the work you are citing even though it mentioned it. The source of the evaporation data is missing.

Response: We have replaced the reference with reference [16].

Line 68:  The cited work for describe the aquifer is a recent study about nitrate in the area, is not the proper cite. You can see “Quaternary Aquifer of the North China Plain—assessing and achieving groundwater resource sustainability” from Foster et al. 2004, and works therein.

Response: We have replaced the reference with the paper that you recommended.

Line 69: a space is missing before the square brackets.

Response: We have added a space.

Line 70-line 78: Cites are missing. What is water-richness?

Response: We have added the reference [17].

Line 79-Line 87: Cites are missing.

Response: We have added the reference [19].

Line 124: A space is missing before the square brackets.

Response: We have added a space.

Line 129: A space is missing before the square brackets.

Response: We have added a space.

Line 132: A space is missing before the square brackets, and in the next sentence is not in the correct order.

Response: We have revised the sentence.

Line 149: I would divide the sentence in two

Response: We have revised " The mean groundwater pH and TDS values were 7.48 and 848.97 mg/L, respectively, of 1.23% of pH and 22.22% of TDS samples exceeded the Grade  III  standard  for  groundwater  quality in  China[13]. " as " The mean groundwater pH and TDS values were 7.48 and 848.97 mg/L, respectively, among which 1.23% of pH and 22.22% of TDS samples exceeded the Grade III standard for groundwater quality in China[19]" in line 174-176.

Line 150: A space is missing before the square brackets.

Response: We have added a space.

Line 171: Correct the sentence

Response: We have revised " Results for calculated WQIs during the dry, rain and transition seasons from are also given in Table 1." as " Results for calculated WQIs during the dry, rain and transition seasons are given in Table 1" in line 198-199.

Line 239: A space is missing before the square brackets.

Line 247: A space is missing before the brackets.

Line 252: A space is missing before the brackets.

Response: We have added a space.

Line 256: Correct the sentence

Response: We have revised "In addition, in dry season, the agricultural chemical fertilizer could not is a mainly pollution source of nitrate in groundwater due to the lack of the driving force of rainfall runoff. " as " In addition, in dry season, the agricultural chemical fertilizer could is not a mainly pollution source of nitrate in groundwater due to the lack of the driving force of rainfall runoff.".

Line 261: Rewrite the sentence, it is grammatically wrong.

Response: We have revised " In the study area, the chemical fertilizers and road salt may not was mainly pollution soures of Cl - . " as " In the study area, the chemical fertilizers and road salt may was not mainly pollution soures of Cl - .".

Line 162: Rewrite the sentence, you use “mainly” three times.

Response: We have revised " This mainly due to the agricultural fertilizer is mainly the nitrogen, phosphorus, potassium, fertilizer, and road salt are mainly used in urban areas, after the snow melts on the road, it directly enters the urban sewage treatment plant through the city's sewage network. " as " This mostly due to the agricultural fertilizer is mainly the nitrogen, phosphorus and potassium in this region, and road salt are primary used in urban areas, after the snow melts on the road, it directly enters the urban sewage treatment plant through the city's sewage network. ".

Line 268: What do you mean with “unscientific discharge”?

Response: We have revised " unscientific discharge " as "untreated discharge" in line 301.

Line 276: A space is missing before the square brackets and another space is missing before the brackets.

Line 278: Spaces are missing after G3, G6, and season.

Line 316: A space is missing before the brackets

Line 317, 318 y 319: spaces are missing after the after the commas and before brackets.

Line 341: A space is missing before the brackets

 Response: We have added a space.

Figure 1: It would be great if the river was included in the figure 1-1. The caption of the figure is incomplete, name figure 1A and 1B and specify in the caption what is display in each figure. It would be very clarifying to have the piezometric head of the area or at least the main recharge areas and the general groundwater flow direction.

 Response: We have revised the Figure 1.

Figure 2: A name or signed to identify each figure is missing. The titles of the graphics and the number in the axis are too small. The legend is the same for the 6 figures, I suggest to make one much bigger to avoid duplicate the information. The caption is heavily incomplete; it should provide information about what is being shown in each figure.

Response: We have revised the Figure 2 according to your sugesstion.

Supplementary Table 2: the values and categories of WQI can be added to table 2 (just adding another column.

Response: We have added the values and categories of WQI to table 2.

Reviewer 2 Report

Here my revision to the proposed manuscript. The work is generally well written with good English and grammar. Only a minor English check is necessary, moreover be careful in follow the MDPI format guidelines.

ABSTRAT

Line 15 change from with “in”

INTRODUCTION

The introduction suffers of a lack in information. The state of art of multivariate statistical analysis is missing. It is important in this case to discuss and show all the previous and recent finding in using PCA, FA and more in the field of geochemistry and hydrology.

Here some papers to read and cite for enhance the introduction:

Zhang, B., Song, X., Zhang, Y., Han, D., Tang, C., Yu, Y., Ma, Y., 2012. Hydrogeochemical characteristics and water quality assessment of surface water and groundwater in Songnen plain, Northeast China. Water Res. 46 (8), 2737e2748.

Busico, G., Cuoco, E., Kazakis, N., Colombani, N., Mastrocicco, M., Tedesco, D., Voudouris, K., 2018. Multivariate statistical analysis to characterize/discriminate between anthropogenic and geogenic trace elements occurrence in the Campania plain, Southern Italy. Environ. Pollut. 234, 260e269

Pereira, H.G., Renca, S., Sataiva, J., 2003. A case study on geochemical anomaly identification through principal component analysis supplementary projection. Appl. Geochem 18, 37e44. https://doi.org/10.1016/S0883-2927(02)00099-9.

Kim, K., Yun, S., Choi, B., Chae, G., Joo, Y., Kim, K., Kim, H., 2009b. Hydrochemical and multivariate statistical interpretations of spatial controls of nitrate concentrations in a shallow alluvial aquifer around oxbow lakes (Osong area, central Korea). J. Cont. Hydrol. 107 (3e4), 114e127. https://doi.org/10.1016/ j.jconhyd.2009.04.007.

Materials and methods

I don’t understand if the figure at line 64 without didascaly is part of figure 1 or not. In any case here my suggestion to modify the image

First, reduce the size of China figure and insert inside the square of figure 1 in the right corner, second add a reference grid outside the figure.

Line 79. If available, please add the % of changes in land use classification in the last 10-20 years.

Line 107: please add the overall precision for anions and cations analysis.

Line 129:  Add and discuss these references for enforce the sentences:

Singh, S., Ghosh, N. C., Gurjar, S., Krishan, G., Kumar, S., & Berwal, P. (2018). Index-based assessment of suitability of water quality for irrigation purpose under Indian conditions. Environmental Monitoring and Assessment, 190, 29–14. https://doi.org/10.1007/s10661-017-6407-3.

Lumb, A., Sharma, T. C., & Bibeault, J. F. (2011). A review of genesis and evolution of water quality index (WQI) and some future directions. Water Quality Exposure and Health, 3(1), 11–24. https://doi.org/10.1007/s12403-011-0040-0.

Line 262-264 please rephase.

Author Response

Here my revision to the proposed manuscript. The work is generally well written with good English and grammar. Only a minor English check is necessary, moreover be careful in follow the MDPI format guidelines.

Response: Thanks for the suggestion and we have revised the flaws according to your request.

General comments:
1.     ABSTRAT

Line 15 change from with “in”

Response: We have revised "from" with “in” in line 17.

    The introduction suffers of a lack in information. The state of art of multivariate statistical analysis is missing. It is important in this case to discuss and show all the previous and recent finding in using PCA, FA and more in the field of geochemistry and hydrology.

Here some papers to read and cite for enhance the introduction:

Zhang, B., Song, X., Zhang, Y., Han, D., Tang, C., Yu, Y., Ma, Y., 2012. Hydrogeochemical characteristics and water quality assessment of surface water and groundwater in Songnen plain, Northeast China. Water Res. 46 (8), 2737e2748.

Busico, G., Cuoco, E., Kazakis, N., Colombani, N., Mastrocicco, M., Tedesco, D., Voudouris, K., 2018. Multivariate statistical analysis to characterize/discriminate between anthropogenic and geogenic trace elements occurrence in the Campania plain, Southern Italy. Environ. Pollut. 234, 260e269

Pereira, H.G., Renca, S., Sataiva, J., 2003. A case study on geochemical anomaly identification through principal component analysis supplementary projection. Appl. Geochem 18, 37e44. https://doi.org/10.1016/S0883-2927(02)00099-9.

Kim, K., Yun, S., Choi, B., Chae, G., Joo, Y., Kim, K., Kim, H., 2009b. Hydrochemical and multivariate statistical interpretations of spatial controls of nitrate concentrations in a shallow alluvial aquifer around oxbow lakes (Osong area, central Korea). J. Cont. Hydrol. 107 (3e4), 114e127. https://doi.org/10.1016/ j.jconhyd.2009.04.007.

Response: We have added these papers in the introduction section.

    Materials and methods

I don’t understand if the figure at line 64 without didascaly is part of figure 1 or not. In any case here my suggestion to modify the image

First, reduce the size of China figure and insert inside the square of figure 1 in the right corner, second add a reference grid outside the figure.

Response: We have revised the figure 1 according to your request.

Line 79. If available, please add the % of changes in land use classification in the last 10-20 years.

Response: We have added the % of changes in land use classification in the last 20 years in line 99-100.

Line 107: please add the overall precision for anions and cations analysis.

Response: We have added the analytical method detection limit values for anions and cations analysis.

Line 129:  Add and discuss these references for enforce the sentences:

Singh, S., Ghosh, N. C., Gurjar, S., Krishan, G., Kumar, S., & Berwal, P. (2018). Index-based assessment of suitability of water quality for irrigation purpose under Indian conditions. Environmental Monitoring and Assessment, 190, 29–14. https://doi.org/10.1007/s10661-017-6407-3.

Lumb, A., Sharma, T. C., & Bibeault, J. F. (2011). A review of genesis and evolution of water quality index (WQI) and some future directions. Water Quality Exposure and Health, 3(1), 11–24. https://doi.org/10.1007/s12403-011-0040-0.

Response: We have added one reference for Lumb, A., Sharma, T. C., & Bibeault, J. F. (2011). A review of genesis and evolution of water quality index (WQI) and some future directions. Water Quality Exposure and Health, 3(1), 11–24.

Line 262-264 please rephase.

Response: We have revised " This mainly due to the agricultural fertilizer is mainly the nitrogen, phosphorus, potassium, fertilizer, and road salt are mainly used in urban areas, after the snow melts on the road, it directly enters the urban sewage treatment plant through the city's sewage network. " as " This mostly due to the agricultural fertilizer is mainly the nitrogen, phosphorus and potassium in this region, and road salt are primary used in urban areas, after the snow melts on the road, it directly enters the urban sewage treatment plant through the city's sewage network. " in line 294-297.

Reviewer 3 Report

Although topic of the manuscript is good and suits well with this journal. Authors need to work a lot on this for make it worth publishing material. Please find few of my comments:

Authors have cited similar works in the same area mentioning hydro-chemical facies of water samples. Similarly, even in the discussion part, you have concluded various things based on the previous works. It is not clear how this work is different from previous one. Authors are suggested to add one paragraph mentioning works done in this area and based on the gaps you found, why this work is important. Also refine the objective Result and discussion part is critically poor. In current status, I got a feeling that authors did not give proper attention before writing any statements/conclusions. Please rewrite this whole section.     Several grammatical errors makes this manuscript very hard to follow at many places. Authors are suggested to carefully check those mistake and rectify it.

For detail comments, please check the reviewed manuscript.

Author Response

Although topic of the manuscript is good and suits well with this journal. Authors need to work a lot on this for make it worth publishing material. Please find few of my comments:

Authors have cited similar works in the same area mentioning hydro-chemical facies of water samples. Similarly, even in the discussion part, you have concluded various things based on the previous works. It is not clear how this work is different from previous one. Authors are suggested to add one paragraph mentioning works done in this area and based on the gaps you found, why this work is important. Also refine the objective Result and discussion part is critically poor. In current status, I got a feeling that authors did not give proper attention before writing any statements/conclusions. Please rewrite this whole section. Several grammatical errors makes this manuscript very hard to follow at many places. Authors are suggested to carefully check those mistake and rectify it.

Response: We have added one paragraph to interpreted the important of this work in line 58-68. We have revised the grammatical errors and rewritten this whole section according to your request.

Detailed comments

It has so many error, please think carefully before writing it again to deliver the clear message you wanted to highlight in line 13.

Response: We have revised " A water quality index and multivariate statistical  techniques  were  used  to  assess groundwater quality and to trace pollution sources in the Hutuo River alluvial-pluvial fan, China. " as "Groundwater quality deterioration has become an environmental problem of widespread concern. In this study, we used a water quality index(WQI) and multivariate statistical techniques to assess groundwater quality and to trace pollution sources in the Hutuo River alluvial-pluvial fan, China" in line 13-16.

As per this sentence, isn't source for groundwater pollution same for both seasons? Please clarify

Response: Yes. The groundwater pollution Source is same in the rainy and transition seasons due to groundwater is affected by rainfall dilution in both seasons.

You can simplify these three objectives in a single one as all three are basically giving same message in line 52-56.

Response: We have simplified these three objectives in a single one according to your suggestion in line 64-68.

Why you wrote rate here in Table 2? What do you mean by rate?

Response: We calculated the ratio of sampling sites to different types of water in order to reflect the overall groundwater quality in this area.

Is this significant number in line 221? Can you please explain what is the standard you have picked based on which you are making such classification?

Response: We define the correlation between the water quality parameters according to Pearson correlation analysis, when p value is less than 0.05, the two indexes show significant correlation, however, when p value is less than 0.01, the two indexes show extremely significant correlation.

How did you come to this conclusion? Did you measure hydraulic parameters of the aquifers and estimated the rate of recharge by rainwater. This is very naive way to conclude in line 223-226.

Response: We have deleted the conclusion.

What is unscientific discharge of domestic sewage in line 268?

Response: We have revised " unscientific discharge " as "untreated discharge" in line 301.

What is the relation between industrial sewage and dry season for high concentration of Fe in dry seasons in line 279?

Response: In this study, the Fe in groundwater could from industrial sewage and the concentration of Fe is higher in dry seasons than in rainy season which may be due to groundwater is not affected by rainfall dilution.

Round 2

Reviewer 1 Report

The authors have performed several modifications several of the main and specific comments and the revised manuscript is easier to read and understand and less redundant than the previous one. However, there are many of the main concerns that have not been resolved yet, as I already pointed out in the first round revision, hydrogeological information is still missing. The possibility of describe the multivariate statistical analysis state of the art and to include the Lumb et al., 2011 reference and to discuss it have been suggested, the authors have included the references suggested but without any further explanation/description of the referenced works. In general, the manuscript some more extra work (structure, figures quality, references supporting the statements and discussion).

References are steel insufficient to support many of the statements, in most cases located studies are used to support general sentences, in other cases references pointed out by the reviewer as not proper references have just been removed from the revised manuscript instead of been replaced by good ones.  

Figures caption explanation are insufficient, and in some cases the legend and axes values fonts are too little to be read. 

There are numerous mistakes in the text, for example in the abstract:

The term TDS is used without previously definition Is it correct to consider water-rock interactions as a pollution?

Author Response

Comments and Suggestions for Authors:

The authors have performed several modifications several of the main and specific comments and the revised manuscript is easier to read and understand and less redundant than the previous one. However, there are many of the main concerns that have not been resolved yet, as I already pointed out in the first round revision, hydrogeological information is still missing. The possibility of describe the multivariate statistical analysis state of the art and to include the Lumb et al., 2011 reference and to discuss it have been suggested, the authors have included the references suggested but without any further explanation/description of the referenced works. In general, the manuscript some more extra work (structure, figures quality, references supporting the statements and discussion).

Response: We have added the some hydrogeological information in figure 1 and we have also described the multivariate statistical analysis state. In addition, due to the reference of the Lumb et al., 2011 was cited in the data analysis section, thus, we did not give further explanation.

References are steel insufficient to support many of the statements, in most cases located studies are used to support general sentences, in other cases references pointed out by the reviewer as not proper references have just been removed from the revised manuscript instead of been replaced by good ones.  

Response: We have replaced the previous references with these recommended by the reviewer and revised some statements.

Figures caption explanation are insufficient, and in some cases the legend and axes values fonts are too little to be read. 

Response: We have revised the figures caption and the axes values fonts.

There are numerous mistakes in the text, for example in the abstract:

Response: We have revised the grammatical mistakes in the text.

The term TDS is used without previously definition Is it correct to consider water-rock interactions as a pollution? 

Response: We have added the definition of TDS before used it. We consider that the water-rock interactions is a pollution source, because it causes an increase in the concentration of HCO3-, TH, Mg2+ and Ca2+ and contaminated groundwater. In this study, groundwater TH had the mean value of 600.32 mg/L, and 83.95% of TH samples surpassed Grade III standard for groundwater quality in China, which is closely related to the water-rock interactions. Therefore, we consider water-rock interactions as a pollution source.

Reviewer 2 Report

I want to congratulate with the authors for the good work done in fulfilled all reviewer requests.

Author Response

Comments and Suggestions for Authors

I want to congratulate with the authors for the good work done in fulfilled all reviewer requests.

Response: Thanks for your suggestion.

Reviewer 3 Report

Looking in to the revised manuscript, I still feel authors should work on especially result and discussion section to support their ideas for the main processes responsible for geo-chemical evolution of the groundwater samples in the study area.

Author Response

Comments and Suggestions for Authors:

Looking in to the revised manuscript, I still feel authors should work on especially result and discussion section to support their ideas for the main processes responsible for geo-chemical evolution of the groundwater samples in the study area.

Response: We have revised some statements in the result and discussion section and added some references to support our ideas.

Round 3

Reviewer 1 Report

The authors have make most of the suggested changes and include some more references.

Reviewer 3 Report

Based on your revision, I recommend for its acceptance in current form.